# Prognostic Significance of p16 and Its Relationship with Human Papillomavirus Status in Patients with Penile Squamous Cell Carcinoma: Results of 5 Years Follow-Up

**DOI:** 10.3390/cancers14246024

**Published:** 2022-12-07

**Authors:** Jad Chahoud, Niki M. Zacharias, Rachel Pham, Wei Qiao, Ming Guo, Xin Lu, Angelita Alaniz, Luis Segarra, Magaly Martinez-Ferrer, Frederico Omar Gleber-Netto, Curtis R. Pickering, Priya Rao, Curtis A. Pettaway

**Affiliations:** 1Department of Genitourinary Oncology, H Lee Moffitt Cancer Center and Research Institute, Tampa, FL 33612, USA; 2Department of Urology, University of Texas MD Anderson Cancer Center, Houston, TX 77030, USA; 3Department of Biostatistics, University of Texas MD Anderson Cancer Center, Houston, TX 77030, USA; 4Department of Pathology, University of Texas MD Anderson Cancer Center, Houston, TX 77030, USA; 5Department of Biological Sciences, University of Notre Dame, Notre Dame, IN 46556, USA; 6Center for Health Promotion and Prevention Research, University of Texas Health Science Center at Houston, Houston, TX 77030, USA; 7Department of Pharmaceutical Sciences, University of Puerto Rico Medical Sciences Campus & Cancer Biology, UPR Comprehensive Cancer Center, San Juan, PR 00936, USA; 8Department of Head and Neck Surgery, University of Texas MD Anderson Cancer Center, Houston, TX 77030, USA; 9Department of Surgery, (Otolaryngology), Yale University School of Medicine, New Haven, CT 06510, USA

**Keywords:** penile squamous cell carcinoma, human papillomavirus, p16, CDKN2A, overall survival, cancer specific survival

## Abstract

**Simple Summary:**

Penile Squamous Cell Carcinoma (PSCC) is a rare but aggressive cancer and approximately 30–50% of cases are associated with high risk human papillomavirus (HR-HPV). HR-HPV infection has been shown to correlate with expression of protein p16^INK4a^ (p16). We wanted to determine if HPV-HR or p16 expression is associated with better outcomes in PSCC patients; therefore, we analyzed 143 patients with a diagnosis of PSCC and available tissue was analyzed for p16^INK4a^ expression and HR-HPV status. Patients with p16+ tumors had a significantly longer median cancer specific survival in comparison to the p16-group (*p* = 0.004), with respective 5-year cancer specific survival probability of 88% (95% CI; 0.84, 1) versus 58% (95% CI; 0.55, 0.76; *p* = 0.004). HPV status did not predict survival outcomes.

**Abstract:**

Penile Squamous Cell Carcinoma (PSCC) is associated with high-risk human papillomavirus (HR-HPV). The immunohistochemical (IHC) test for p16^INK4a^ (p16) is highly correlated with HR-HPV expression in other SCCs. To investigate whether the expression of p16 IHC or HR-HPV is associated with survival in PSCC, we conducted a single institution analysis of 143 patients with a diagnosis of PSCC and, available tissue were tested for p16 IHC staining patterns, histological subtype, tumor grade, and lymphovascular invasion (LVI) by an experienced pathologist. HR-HPV status using the Cobas PCR Assay or the RNAScope high-risk HPV in situ hybridization kit were also assessed. Patient characteristics were summarized using descriptive statistics of clinico-pathologic variables. Kaplan–Meier was used to estimate median overall survival (OS), cancer specific survival (CSS) and correlated with HPV, p16, and other study variables. Patients with p16+ tumors had a significantly longer median CSS in comparison to the p16– group (*p* = 0.004), with respective 5-year CSS probability of 88% (95% CI; 0.84, 1) versus 58% (95% CI; 0.55, 0.76; *p* = 0.004). HPV status did not predict survival outcomes. Multivariable analysis with respect to OS and CSS, showed that p16+ status was associated with a lower risk of death (HR = 0.36, 95%CI; 0.20–0.67, *p* = 0.001), and improved CSS (HR = 0.20, 95% CI; 0.07–0.54, *p* = 0.002) after adjusting for covariates. In conclusion, tumor p16 status via IHC was an easy to perform independent prognostic factor for OS and CSS that correlates with HR-HPV expression.

## 1. Introduction

A major etiologic pathway of penile squamous carcinoma (PSCC) involves high-risk human papillomavirus (HPV). HPV infection occurs in about 30–50% of penile cancers but varies significantly via histologic subtype [1,2,3]. There are >150 various HPV genotypes, however only a few (HPV 16, 18, 31, 33, 45, 52) have been shown to be associated with PSCC with HPV 16 being the most prevalent genotype [1,2,3]. These high risk HPV genotypes are thought to be involved in the carcinogenesis of penile cancer through the activity of viral E6 and E7 oncoproteins that are capable of binding to and inactivating the p53 and the retinoblastoma-1 tumor suppressor proteins (Rb), respectively [4]. E7’s inhibition of the Rb pathway leads to increased expression of the p16^INK4a^ (CDKN2A) protein (also known as p16). Overexpression of p16 can be detected by immunohistochemical (IHC) staining and has been found to be a reliable marker for high-risk HPV infection in oropharyngeal SCC and to a lesser extent in PSCC [5,6,7,8,9]. This study aims to provide additional evidence regarding the prognostic implications of assessing HPV and p16 status in a cohort of men with PSCC and 5-year follow-up.

## 2. Materials and Methods

### 2.1. Patient Population and Study Design

We conducted a single institution analysis of patients with PSCC who received treatment at The University of Texas, MD Anderson Cancer Center (MDACC) between 1991 and 2017. Eligible patients were defined as those with PSCC, determined by pathology review (PR), and had tissue available for analysis. We conducted p16 immunohistochemical staining (IHC) and HPV polymerase chain reaction (PCR) testing. The demographic, clinical, and treatment data for each patient were obtained from retrospective chart review from the electronic medical records (Appendix A). Patients were considered to be pathologically node-negative if they were clinically node negative and remained relapse-free during follow-up. Histopathology results are reported according to the American joint Committee on Cancer (AJCC) 8th edition tumor, node, and metastases classification for PSCC [10,11].

### 2.2. Protocols for HPV Detection 

HPV detection was performed using Cobas PCR HPV assay on all samples. However, in a subset of samples (44) Cobas provided an invalid reading and the high-risk HPV in situ hybridization assay (HPV-ISH) was used as an alternative strategy on these samples to clarify HPV status. Cobas HPV testing (Roche Diagnostics, Indianapolis, IN, USA) was performed in a similar manner to published methods [12] using formalin fixed paraffin embedded (FFPE) tissue sections. In each testing run, positive and negative controls were included. The testing results were reported as HPV 16 and 18 genotypes or collectively as “other high risk” that could include any of 12 other high-risk HPV types (HPV 31, 33, 35, 39, 45, 51, 52, 56, 58, 59, 66 and 68). The HPV-ISH hybridization assay uses unstained FFPE histology slides and the RNAscope^®^ 2.5 LS Probe-HPV HR8 kit (511628, ACD Biotech, Palo Alto, CA, USA) with an automated Leica BOND-III RNA-scope system allowing it to be used in routine clinical care. The pooled probe can recognize high risk HPV genotypes 16, 18, 31, 33, 35, 45, 52, and 58 as a “pooled result” as either negative or positive but does not provide the specific genotype. In each run, appropriate controls for the RNAscope 2.5 LS probe (300038, ACD Biotech, Abingdon, UK) were used. 

### 2.3. Immunohistochemistry for p16

Histological subtype, tumor grade, LVI (lympovascular invasion), and p16 staining patterns were confirmed by a genitourinary pathologist (PR). For p16 IHC, all samples were tested, and stained on a BenchMark Autostainer (Ventana Medical Systems, Tucson, AZ, USA) as described by the manufacturer’s protocol using a prediluted mouse monoclonal antibody (CINtec^®^ p16 Histology, clone E6H4, Ventana Medical Systems, Tucson, AZ, USA). p16 staining patterns of 0, 1, 2, and 3 were classified using previously described categories [13,14]. Based on previously published methods [5,13,14,15] and our own observations, we created a hybrid system (HS) for analyzing p16 via IHC (Table 1). The HS method used both the percent (>75%) positive staining of the tumor section and the staining pattern. For representative images, slides were scanned using Aperio AT2 total slide scanner (AT2, Leica Biosystems, Vista, CA, USA) and images are at 2X amplification. 

### 2.4. Statistical Analyses 

The distribution of each continuous variable was summarized by its mean, standard deviation, and range. Categorical variables were summarized in terms of their frequencies and percentages. The associations between the binary outcomes and other factors were assessed using Fisher’s Exact test or Wilcoxon Rank sum test as appropriate. Kappa coefficients were used to evaluate the agreement between different p16 definitions and HPV status. The primary endpoints included CSS and OS. CSS was defined as the time from diagnosis until the time of death related to PSCC or censored at the last follow-up. OS was defined as the time from diagnosis until the time of death or censored at the last follow-up. The last follow-up date was defined as the last time the patient was seen by the clinical team within our health care system or the last time the patient corresponded with the hospital to update their disease status if they were being followed by outside providers. We estimated the survival outcomes for the overall cohort using the Kaplan–Meier method, comparing the survival outcomes among subgroups of patients by p16 and HPV status using logrank test. The Cox regression proportional hazard model was used for multivariate analysis, predicting OS and CSS based on the backward model selection. The variables that showed potential significant effect on OS and CSS in univariable models were considered in the initial multivariable model. *p* Values < 0.05 were considered statistically significant. SAS software 9.4 (SAS Institute, Cary, NC, USA) was used for statistical analysis. Abbreviations for 95% confidence interval (95% CI) or interquartile range (IQR) used throughout the text. 

## 3. Results

### 3.1. Patients and Disease Characteristics

One hundred forty-three patients were identified. Median age at diagnosis for this cohort was 58.9 years (IQR, 48.6–68.6 years) and the median time between onset of symptoms and date of diagnosis was 6.5 months (IQR, 2.03–21.53). Patient characteristics are summarized in Appendix A. The cohort was multiracial and multiethnic. Established risk factors for PSCC were present in approximately 43–72% of study subjects. High risk primary tumors were noted in 102 patients (73.4%) and 69 patients (49.3%) with LVI. Seventy-one patients (51.1%) had clinically N0 disease at the time of diagnosis. Nodal surgical intervention was not performed in 24 patients, with 10 patients having lymph node biopsies. One hundred seventeen patients (81.8%) underwent nodal surgical interventions with median of 28 (IQR, 20–39) total collected lymph nodes and median 1 (IQR, 1–2) lymph node(s) with metastatic PSCC (LN+). Subsequent to surgical treatment 70 patients (48.9%) with sufficient follow up were pN+ with 40 patients having evidence of extra nodal extension (i.e., pN3) on pathology review. Forty-six patients received neoadjuvant chemotherapy, the majority (*n* = 32, 97%) received the combination chemotherapy regimen with cisplatin, ifosfamide and paclitaxel (TIP). The median follow-up time for the study was 5.8 years (95% CI; 4.7, 7.5 yr), during this period, 70 (49.7%) patients died, 42 (29.8%) patients died from PSCC, 11 (7.8%) patients were alive with disease, and 60 (42.6%) patients remained alive without evidence of disease. 

### 3.2. HPV and p16 Status

Forty-seven patients (32.9%) were positive for HR-HPV with 45 (31.5%) positive for p16 using our HS IHC analysis method (Appendix A). Our HS method uses both the staining pattern and the percent staining in the tumor in assessing p16 expression (Table 1). We observed that all samples with a pattern of 0 or 1 had less than 75% staining (Figure 1A,B). While all samples with a pattern of 3 had more than 75% staining (Figure 1E). All samples with a staining pattern of 2 and over 75% staining were considered indicative of p16 overexpression (Figure 1D, Table 1) while samples with a staining pattern of 2 and less than 75% staining were considered negative for p16 overexpression (Figure 1C, Table 1). Using our HS analysis method, there were 20 samples where HPV and p16 status were discordant (Appendix A). To determine the overall agreement between HPV and p16 status, the kappa coefficient was determined based on our HS method or the staining pattern method (where only staining pattern 3 is considered p16 positive). For the HS analysis method, the kappa coefficient with standard error was 0.680 ± 0.066 (95% CI; 0.550, 0.809) while the staining pattern 3 cut off was 0.603 ± 0.072 (95% CI; 0.461, 0.745). The higher kappa coefficient with HS analysis indicates that we have better agreement between HPV and p16 status using the HS method versus staining pattern alone. In our cohort, the difference in the p16 status between the HS method and the staining pattern method was found only for samples with a staining pattern of 2. In our cohort, a total of 17 samples had a staining pattern of 2 (Appendix A) and using our HS analysis only 6 of these samples were discordant (Sample ID 24, 29, 65, 74, 93, 137). However, if the staining pattern was used alone, with a pattern of 2 considered p16 negative, 9 samples (Sample ID 1, 30, 58, 68, 88, 105, 111, 148, 137) would have been discordant (Appendix A). 

### 3.3. HPV and p16 Expression Status and Clinicopathological Findings

The association between HPV and p16 status and clinicopathological features are shown in Appendix A, respectively. The Hispanic cohort tended to have HPV/p16 negative tumors (over 80% HPV/p16 negative). The presence of phimosis was also correlated with HPV/p16 negative tumors (75–80%). In contrast Basaloid tumors were overwhelmingly HPV/p16 positive (80–93%). Grade 1–2 tumors were more commonly HPV and p16 negative and grade 3 tumors were more evenly distributed between HPV and p16 positive and negative status. There was no correlation between most of the patient characteristics and HPV/p16 status including both primary tumor and lymph node stage. 

### 3.4. HPV and p16 Status and Survival in PSCC

To determine how our p16 HS analysis compared to the previously published staining pattern 3 IHC cut off (Figure 1E, [13,14]), the prognostic ability of p16 status in CSS and OS was determined using both methods. When our HS analysis is used to determine p16 status, p16 status was significant in CSS with a hazard ratio (HR) of 0.30 (95% CI; 0.13, 0.72) with a *p*-value of 0.007 while p16 status showed non-significant impact on OS with HR = 0.60 (95% CI; 0.35, 1.04) with a *p*-value of 0.068. Using staining pattern 3 as p16 positive, p16 status was significant in CSS with HR = 0.32 (95% CI; 0.11, 0.90) and *p*-value of 0.03 while p16 status showed non-significant impact on OS with HR = 0.63 (95% CI; 0.34, 1.17) and *p*-value of 0.145. Therefore, p16 status is significant in CSS using either our HS analysis or the staining pattern method. However, we do observe lower *p*-values with our HS analysis method and a better kappa coefficient therefore, p16 status based on our HS method was used in all other analyses. 

Univariate Cox analysis for OS and CSS were performed (Table 2 and Table 3). Statistical significance was found for CSS based on p16 status (*p* = 0.004, CSS, Table 3) but not OS (*p* = 0.065, OS, Table 2). No statistical significance in OS or CSS was noted based on HPV status. Patients with p16+ tumors showed a significantly longer median CSS in comparison to the p16– group, with respective 5-year CSS probability of 88% (95% CI; 0.77, 1) in comparison to 58% (95% CI; 0.48, 0.71). In contrast, HPV status did not predict survival outcomes, with 5-year CSS probability for HPV (+) of 76% (95% CI; 0.63, 0.92) compared to 64% (95% CI; 0.54, 0.75) for HPV (−) patients with a *p* value of 0.14. Other variables consistently associated with both OS and CSS included both clinical and pathologic primary tumor and nodal stage (Table 2 and Table 3).

Kaplan–Meier (KM) plots for OS and CSS stratified by p16 and HPV status are shown in Figure 2. p16 status had a significant impact on CSS, while HPV status had no significant impact on either variable. We determined if p16 or HPV status had any significant impact on OS or CSS in the subset of patients with advanced disease as demonstrated by pathologic tumor stage pT ≥ 3 versus ≤ 2 (31 patients in total) and patients with nodal stage pN ≥ 2 versus ≤ 1 (52 patients in total). KM plots revealed that p16 status had a significant impact on OS and CSS among patients with high pT stage (Figure 3A–D) and impacted CSS for patients with pN ≥ 2 stage involvement (Figure 3E–H). Among the advanced disease cohort HPV positive status was also correlated with improved OS among patients with advanced stage primary tumors. However, for patients with both ≤ pT2 stage and ≤ pN1 stage, no significant survival difference was found between p16 positive vs. negative or HPV positive vs. negative cohorts. 

### 3.5. Multivariable Models for OS and CSS

Table 4 and Table 5 provide the multivariable models for OS and CSS based on a backward model selection. After adjusting for other clinical factors, patients that were p16 positive had a lower hazard of death overall 0.36 (95% CI: 0.20, 0.67) compared to p16 negative patients (Table 4). Other variables correlated with OS included nodal stage and tumor grade. As shown in Table 5, p16 status was an important novel independent predictor of CSS along with grade and advanced lymph node burden. 

## 4. Discussion

PSCC is a rare but lethal cancer whose biology is driven, at least in part, by differences in high risk HPV status [16,17,18]. Subsequent to high risk HPV infection, the p16^INK4a^ protein (p16) is upregulated and has been detected in high risk HPV related malignancies utilizing IHC [2,5,6,8,9,17]. The p16 protein is not only a surrogate for HPV expression but has become a reproducible, reliable, and cost-efficient predictor of prognosis following a cancer diagnosis in oropharyngeal (OPC) SCC [8,19,20]. The correlation between p16 and HPV has been studied in a more limited fashion in PSCC and a similar strong correlation to that of OPSCC was observed (88%) [6,9]. To validate this correlation in PSCC, we investigated the correlation between p16 and HPV and the prognostic significance of p16 in a larger cohort of patients with PSCC and median 5.8-year (95% Cl; 4.7, 7.5) follow-up.

The expression of p16 was found to be superior to all other clinical, HPV genotyping, or IHC parameters for predicting prognosis for patients with OPSCC [21,22]. We found a similar prognostic effect with p16+ PSCC in our cohort. Patients with p16+ tumors exhibited an improvement in 3-year and 5-year CSS, respectively (92% vs. 65% and 88 vs. 58%; *p* = 0.004). These findings were clinically relevant and feasible to document utilizing IHC and a binary cut off value (i.e., >75%) for p16 status. Of note, p16 positivity may exert its prognostic significance among patients with more advanced PSCC (i.e., pT ≥ 3, pN ≥ 2) as the correlation with better survival outcomes was significant among those with advanced versus lower stage disease (pT ≤ 2, pN ≤ 1). Among advanced disease PSCC patients the use of multimodality therapy is certainly more prevalent. It is plausible but not proven that p16 as a potential marker of HPV driven malignancy may correlate with response to multimodal therapies such as chemoradiation or chemotherapy. In one recent study patients with advanced PSCC that were HPV+ exhibited significantly greater overall survival when treated with radiotherapy when compared with the HPV− cohort [23].

Our study highlights the importance of p16 staining in PSCC and how this relatively simple methodology using IHC compares to the more costly HPV testing technologies. To our surprise HPV status was not shown to have the same prognostic ability as p16 staining in the overall cohort but was significantly associated with an overall survival benefit among patients with advanced primary tumor stage. A recent meta-analysis by Sand and colleagues of PSCC studies [24] that assessed HPV and p16 expression in a cohort of 649 men that included 20 studies aligns with our findings. They found that p16 was associated with positive prognostic value for CSS with a hazard ratio of 0.45 (95% CI; 0.30 to 0.69) as was high risk HPV expression but with a smaller protective effect (hazard ratio = 0.61, 95% CI; 0.38 to 0.98). The observation that high risk HPV expression did not exhibit the same predictive effect as p16 in our cohort may be explained by p16′s downstream role in cell cycle control, and that p16 may serve as a biomarker for the degree that the tumor is HPV driven in PSCC. However, caution is advised as some OPSCC studies have found high p16 expression in a small subset of tumors where HPV-DNA was not detected [25,26]. Interestingly, these patients with high p16 expression and undetected HPV-DNA had poorer survival curves similar to alcohol and tobacco related OPSCC and not HPV-driven OPSCC. Whether the p16+, HPV negative phenotype has a better or poorer prognosis than the concordant phenotype clearly deserves further study in PSCC. 

In our 143 patient samples, HPV and p16 status were discordant in 20 samples. All HPV testing was initially run using the Cobas system and a subset that gave invalid results [27] were analyzed using the alternative RNAscope analysis. These two assays are significantly different. Cobas is an automated system that extracts DNA from FFPE tissue slices and uses real-time PCR amplification of the L1 gene from 14 high-risk HPV genotypes [28], while RNAscope is an in situ hybridization technique that stains E6 and E7 mRNA of high-risk HPV genotypes directly within FFPE tissue sections [29]. However, in our small sample set we observed no significant difference in discordance between HPV and p16 status utilizing either Cobas or RNAscope assays. Thus, it is unlikely that the assays used resulted in the discordance noted. 

A novel finding of our study is that the use of a binary cut off of 75% staining was a simple method to assess p16 positive or negative status. A stronger overall agreement between HPV and p16 status was found using our HS method versus the staining pattern method as measured by the kappa coeffecient. Considering its prognostic value we noted a stronger correlation with CSS (HR = 0.30; 95% CI; 0.13, 0.72; *p*-value 0.007) utilizing our HS method over a previously published staining pattern method (HR = 0.32; 95% CI; 0.11, 0.90; *p*-value 0.03) [13,14]. However, we acknowledge the lack of independent validation of our p16 HS method and evaluation of p16 staining by a single genitourinary pathologist. While we showed that p16 IHC with a simple 75% positive cut off to categorize cases at our center was a strong prognostic factor for survival we invite further validation from other experienced centers treating penile cancer patients. In this manner p16 staining determined by multiple pathologists to determine inter-observer concordance and prognostic capability would be optimal. Of note, such a validation study going forward will likely be feasible as the World Health Organization recently recommended that p16 status be reported as a “requirement” for PSCC pathology reporting [30]. 

The primary limitations of this study are its retrospective nature in the setting of a tertiary referral center and heterogeneous treatment regimens. To minimize these effects, we analyzed the importance of a variety of known prognostic covariates on survival outcome. Clinical and pathologic correlates of advanced disease were significant predictors of both OS and CSS. However, after adjusting for all variables, p16 remained an important independent predictor of both OS and CSS. Thus, the present study provides further evidence that p16 IHC is a rapid, easy to perform test that has a high sensitivity and high negative predictive value for assessing HPV status in PSCC. As such p16 status serves as an important prognostic marker and excellent first line test to potentially describe the biology driving a given patient’s PSCC. 

## 5. Conclusions

We validate p16 status as a valuable independent survival biomarker in PSCC that correlates with HPV driven malignancy.

## Figures and Tables

**Figure 1 cancers-14-06024-f001:**
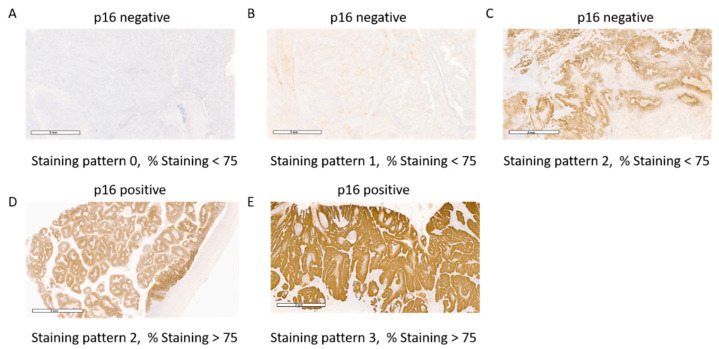
Representative p16 IHC images of PSCC tissue illustrating the staining pattern and the % staining. In our HS-system, staining pattern 0 (**A**), 1 (**B**), and 2 (**C**) with <75% staining are considered p16 negative. Only staining pattern 2 with >75% staining (**D**) and staining pattern 3 with >75% staining (**E**) were considered p16 positive.

**Figure 2 cancers-14-06024-f002:**
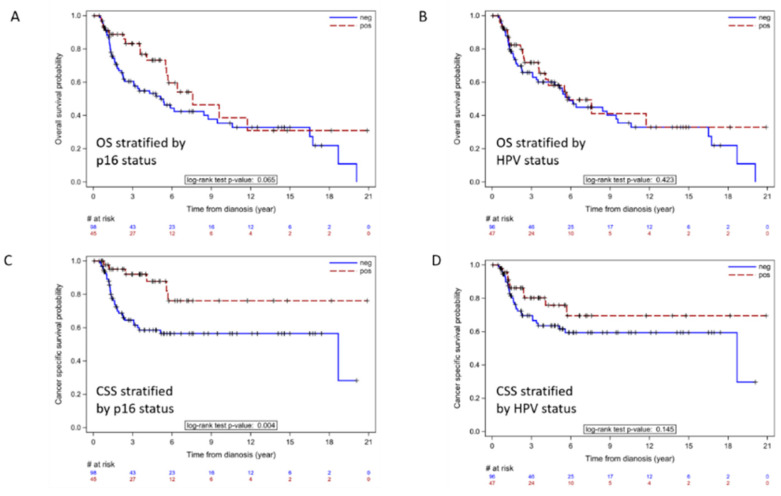
KM plots for OS and CSS stratified by p16 (**A**,**C**) or by HPV status (**B**,**D**). p16 positive patients have better overall CSS (*p* = 0.004) than p16 negative patients.

**Figure 3 cancers-14-06024-f003:**
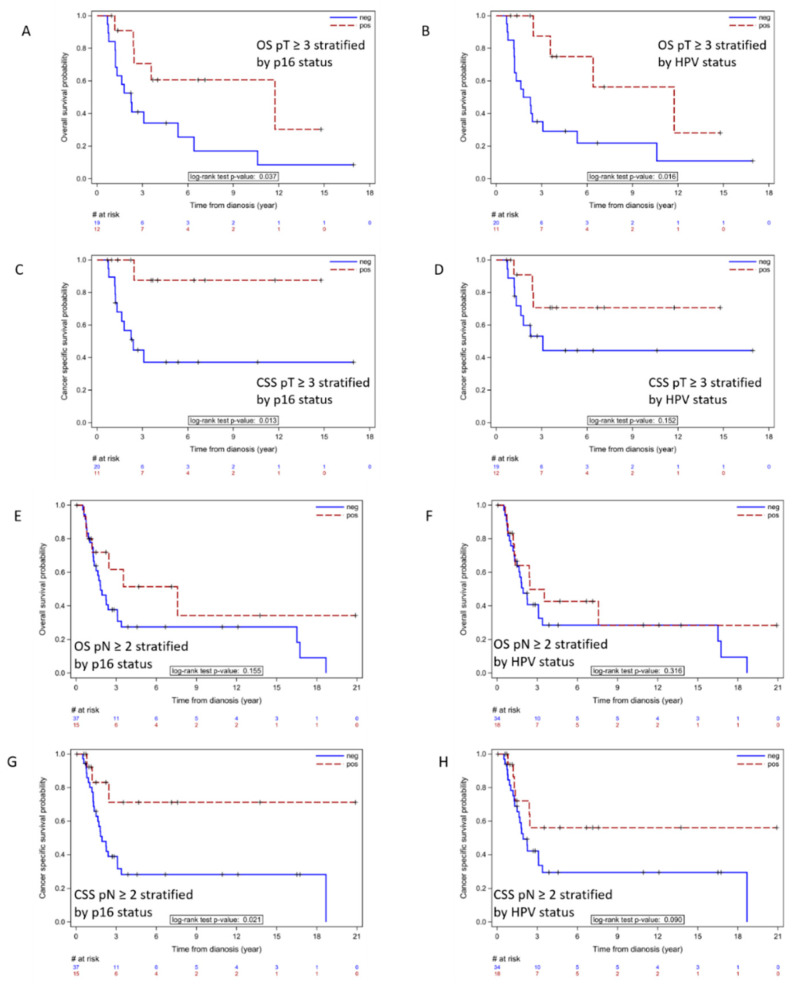
KM plots for CSS and OS for patients with pT stage of 3 and 4 stratified by p16 status (**A**,**C**) or by HPV status (**B**,**D**). Similar KM plots were generated for patients with 2 to 3 level of nodal involvement stratified by p16 (**E**,**G**) or by HPV status (**F**,**H**).

**Table 1 cancers-14-06024-t001:** Hybrid system (HS) for analyzing p16 IHC. A combination of the staining pattern [13,14] and the percent staining was used to determine if p16 status was positive or negative. When more than 1 pattern was observed in the same tumor, the highest one was considered as representative of the whole lesion.

p16 Status	Staining Pattern Number	Definition of Staining Pattern	% Staining
Negative	0	complete absence of p16 staining in all neoplastic cells	0%
Negative	1	spotty, patchy, and discontinuous individual staining in some neoplastic cells	<75%
Negative	2	extensive discontinuous staining pattern with small clusters of positive neoplastic cells	<75%
Positive	2	extensive discontinuous staining pattern with small clusters of positive neoplastic cells	>75%
Positive	3	entire and continuous cytoplasmic or nuclear staining in all neoplastic cells	>75%

**Table 2 cancers-14-06024-t002:** Univariate Cox Analysis for OS at 12 months (1 yr), 36 months (3 yr) and 60 months (5 yr). HPV and p16 status and other significant patient characteristics shown.

Univariate for OS	Level	N	Event	Rate at 12 Months (95%CI)	Rate at 36 Months (95%CI)	Rate at 60 Months (95%CI)	*p*-Value
	All patients	143	70	0.91 (0.86, 0.96)	0.68 (0.6, 0.77)	0.59 (0.5, 0.68)	
HPV	neg	96	50	0.9 (0.84, 0.96)	0.66 (0.56, 0.77)	0.58 (0.48, 0.7)	0.423
	pos	47	20	0.91 (0.84, 1)	0.72 (0.59, 0.87)	0.58 (0.44, 0.77)	
p16	neg	98	53	0.9 (0.85, 0.97)	0.61 (0.51, 0.72)	0.52 (0.42, 0.64)	0.065
	pos	45	17	0.91 (0.83, 1)	0.83 (0.73, 0.96)	0.73 (0.6, 0.89)	
cT stage	1	45	20	0.93 (0.86, 1)	0.81 (0.7, 0.94)	0.72 (0.59, 0.88)	0.002
	2	78	38	0.91 (0.84, 0.97)	0.64 (0.53, 0.77)	0.51 (0.39, 0.66)	
	3	16	8	0.88 (0.73, 1)	0.68 (0.48, 0.96)	0.68 (0.48, 0.96)	
	4	3	3	1 (1, 1)			
cN stage	0	74	28	0.94 (0.89, 1)	0.85 (0.77, 0.94)	0.71 (0.59, 0.84)	<0.001
	1	28	17	0.96 (0.89, 1)	0.63 (0.47, 0.86)	0.59 (0.42, 0.82)	
	2	12	10	0.75 (0.54, 1)	0.25 (0.09, 0.67)		
	3	29	15	0.82 (0.69, 0.98)	0.47 (0.3, 0.73)	0.47 (0.3, 0.73)	
pT stage	1	37	16	0.97 (0.92, 1)	0.83 (0.71, 0.96)	0.75 (0.61, 0.92)	<0.001
	2	71	32	0.9 (0.83, 0.97)	0.69 (0.58, 0.82)	0.58 (0.46, 0.73)	
	3	26	15	0.88 (0.77, 1)	0.63 (0.46, 0.86)	0.53 (0.36, 0.78)	
	4	5	5	1 (1, 1)			
pN stage	0	71	24	0.99 (0.96, 1)	0.87 (0.79, 0.96)	0.76 (0.65, 0.89)	0.001
	1	16	9	0.93 (0.82, 1)	0.73 (0.54, 1)	0.73 (0.54, 1)	
	2	12	5	1 (1, 1)	0.56 (0.31, 1)	0.42 (0.18, 0.94)	
	3	40	30	0.75 (0.63, 0.9)	0.41 (0.28, 0.6)	0.31 (0.19, 0.51)	
Grade	1	27	7	0.96 (0.89, 1)	0.82 (0.68, 1)	0.82 (0.68, 1)	0.017
	2	55	29	0.94 (0.88, 1)	0.65 (0.53, 0.8)	0.57 (0.44, 0.74)	
	3	61	34	0.85 (0.76, 0.95)	0.64 (0.52, 0.78)	0.5 (0.37, 0.66)	
LVI	0	71	29	0.93 (0.86, 0.99)	0.71 (0.6, 0.83)	0.62 (0.5, 0.76)	0.043
	1	69	40	0.88 (0.81, 0.96)	0.64 (0.53, 0.77)	0.53 (0.42, 0.68)	
Number of Positive Nodes	0	48	17	1 (1, 1)	0.86 (0.76, 0.97)	0.72 (0.58, 0.89)	<0.001
	1–2	43	22	0.9 (0.82, 1)	0.7 (0.57, 0.86)	0.64 (0.5, 0.81)	
	>2	27	21	0.81 (0.67, 0.97)	0.35 (0.2, 0.6)	0.2 (0.09, 0.46)	

**Table 3 cancers-14-06024-t003:** Univariate Cox Analysis for CSS (Table 3) at 12 months (1 yr), 36 months (3 yr) and 60 months (5 yr). HPV and p16 status and other significant patient characteristics shown.

Univariate for CSS	Level	N	Event	Rate at 12 Months (95%CI)	Rate at 36 Months (95%CI)	Rate at 60 Months (95%CI)	*p*-Value
	All patients	143	42	224.2 (224.2, NA)	0.93 (0.89, 0.98)	0.73 (0.66, 0.82)	
HPV Status	neg	96	32	0.92 (0.87, 0.98)	0.7 (0.6, 0.8)	0.64 (0.54, 0.75)	0.145
	pos	47	10	0.96 (0.9, 1)	0.8 (0.69, 0.94)	0.76 (0.63, 0.92)	
p16 Status	neg	98	36	0.91 (0.86, 0.97)	0.65 (0.55, 0.76)	0.58 (0.48, 0.71)	0.004
	pos	45	6	0.98 (0.93, 1)	0.92 (0.84, 1)	0.88 (0.77, 1)	
cT stage	1	45	7	1 (1, 1)	0.9 (0.81, 1)	0.86 (0.76, 0.98)	<0.001
	2	78	28	0.92 (0.86, 0.98)	0.66 (0.56, 0.79)	0.57 0.46, 0.72)	
	3	16	3	0.88 (0.73, 1)	0.8 (0.61, 1)	0.8 (0.61, 1)	
	4	3	3	1 (1, 1)			
cN stage	0	74	12	0.97 (0.93, 1)	0.91 (0.84, 0.98)	0.84 (0.75, 0.95)	<0.001
	1	28	8	0.96 (0.89, 1)	0.71 (0.55, 0.92)	0.66 (0.49, 0.89)	
	2	12	9	0.83 (0.65, 1)	0.28 (0.11, 0.72)		
	3	29	13	0.85 (0.73, 1)	0.49 (0.32, 0.76)	0.49 (0.32, 0.76)	
pT stage	1	37	7	1 (1, 1)	0.85 (0.73, 0.98)	0.85 (0.73, 0.98)	<0.001
	2	71	21	0.92 (0.86, 0.99)	0.75 (0.65, 0.87)	0.65 (0.53, 0.8)	
	3	26	7	0.92 (0.82, 1)	0.73 (0.57, 0.94)	0.67 (0.5, 0.91)	
	4	5	5	1 (1, 1)			
pN stage	0	71	7	1 (1, 1)	0.92 (0.85, 0.99)	0.87 (0.79, 0.97)	<0.001
	1	16	6	0.93 (0.82, 1)	0.8 (0.62, 1)	0.8 (0.62, 1)	
	2	12	5	1 (1, 1)	0.56 (0.31, 1)	0.42 (0.18, 0.94)	
	3	40	23	0.81 (0.7, 0.95)	0.44 (0.3, 0.64)	0.37 (0.24, 0.58)	
Grade	1	27	2	1 (1, 1)	0.9 (0.78, 1)	0.9 (0.78, 1)	0.018
	2	55	18	0.94 (0.88, 1)	0.71 (0.59, 0.85)	0.65 (0.53, 0.81)	
	3	61	22	0.9 (0.82, 0.98)	0.68 (0.56, 0.82)	0.6 (0.47, 0.76)	
Number of Positive Nodes	0	48	6	1 (1, 1)	0.88 (0.79, 0.99)	0.84 (0.73, 0.9)	<0.001
	1–2	43	13	0.95 (0.88, 1)	0.75 (0.62, 0.91)	0.72 (0.59, 0.89)	
	>2	27	19	0.84 (0.71, 1)	0.36 (0.21, 0.63)	0.21 (0.09, 0.48)	

**Table 4 cancers-14-06024-t004:** Multivariable Model for OS. p16 status is statistically significant in model.

Multivariable Model for OS		Hazard Ratio (95% Confidence Interval)	*p*-Value
p16 status	Pos vs. Neg	0.36 (0.20, 0.67)	0.001
cN stage	0 vs. 3	0.56 (0.28, 1.14)	0.110
	1 vs. 3	0.94 (0.44, 2.00)	0.868
	2 vs. 3	2.94 (1.22, 7.07)	0.016
Age at Diagnosis		1.02 (1.00, 1.04)	0.073
Grade	1 vs. 3	0.23 (0.10, 0.55)	0.001
	2 vs. 3	0.69 (0.40, 1.19)	0.179

**Table 5 cancers-14-06024-t005:** Multivariable Model for CSS. p16 status is statistically significant in model.

Multivariable Model for CSS		Hazard Ratio (95% Confident Interval)	*p*-Value
p16 status	Pos vs. Neg	0.20 (0.07, 0.54)	0.002
Grade	1 vs. 3	0.20 (0.05, 0.92)	0.039
	2 vs. 3	0.42 (0.20, 0.87)	0.019
Number of Positive Nodes	1–2 vs. 0	2.45 (0.93, 6.47)	0.071
	>2 vs. 0	7.52 (2.81, 20.10)	<0.001

## Data Availability

The data presented in this study are available on request from the corresponding author.

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
