# Peer review of "Prognostic Significance of p16 and Its Relationship with Human Papillomavirus Status in Patients with Penile Squamous Cell Carcinoma: Results of 5 Years Follow-Up"

_cancers, 2022, doi:10.3390/cancers14246024_

Round 1

Reviewer 1 Report

I congratulate the authors for this important article.

The whole topic is well investigated and analyzed. The article is well written.

There are few observations/recommendations:

1-     The study is based on new evaluation of the p16 IHC with new scheme and cutoff of 75%. How will be the overall results regarding the OS and CSS if the usual assessment for p16 -IHC (p16-IHC is positive only with diffuse staining- pattern 3) is used?

2-     The evaluation of p16 IHC with this new scheme is done by a single pathologist. This can lead to subjective bias. It is recommended that the inter-observer concordance for the evaluation of the p16 IHC with this new scheme should be examined by other pathologists. Alternatively, this should be mentioned in the discussion as a limitation of the study.

3-     The discordance between p16 status and HR-HPV status needs more discussion. (according to the statistics in this study: 20 discordant cases out of a total of 143: 9 cases with p16+/HPV- and 11 cases with p16-/HPV+). Which influence do the other 150 HPV types have?

4-     Section 4, lines 272-274: The discordance between the results of p16 and HR-HPV needs more discussion. Line 274 is particularly unconvincing for this discordance.

5-     Section 3.2 , lines 163-167: the statistics and the wording in these lines are difficult to understand. The same statistics cannot be obtained from the accompanying table (S2).

6-     Section 3.4 , lines 183-184: OS with P=0,065 > 0,05. This is statistically not significant. Therefore, only the univariate Cox analysis for CSS is statistically significant.

7-     Section 3.4 , lines 211-212 and Section 4, lines 263-264: “HPV positive status trended toward significance among patients with advanced nodal disease”. In the corresponding figure (3H) stands log-rank test with p-value: 0.090 >0,05 , this is statistically not significant. Trending is not scientific or statistical proof.

8-     Abbreviations must be clarified e.g. LVI, CI, IQR.

9-     Conclusion for this good study can be improved.

10-  In the title: Results at 5 years. Alternatively: Results of 5 years follow-up.

Best regards

Reviewer 2 Report

The paper by Chahoud et al is interesting trying to validate the correlation between p16 and HPV and the prognostic significance of  p16 in a relatively  large  cohort of patients with PSCC.  Main conclusion is that   p16 status is a valuable independent survival biomarker in PSCC that  correlates with HPV driven malignancy.

Although I’m not a statistician, analyses seem to be well conducted and results well described. However, I have a major concern about employed diagnostic methodologies.  This porblem must be fully addressed before publication of this paper. HPV detection was done with two different technologies, COBAS that detects L1 region of HPV and RNAscope that detects viral mRNA of E6/E7 genes. The last methodology is less sensitive but more specific detecting HPV that is transcriptionally active. Moreover, L1 gene detected by COBAS, in some tumors, can be deleted during viral integration. Therefore I believe that all sample should be analyzed with same technology.

In alternative, data on concordance between p16 and HPV should be presented in relationship with COBAS or RNAscope analysis.

This possible bias of the performed HPV analysis should be fully discussed in a paragraph of discussion.

Round 2

Reviewer 2 Report

Paper was substantially improved and can be published.